# Evidence for Adaptive Selection in the Mitogenome of a Mesoparasitic Monogenean Flatworm *Enterogyrus malmbergi*

**DOI:** 10.3390/genes10110863

**Published:** 2019-10-30

**Authors:** Dong Zhang, Hong Zou, Shan G. Wu, Ming Li, Ivan Jakovlić, Jin Zhang, Rong Chen, Wen X. Li, Gui T. Wang

**Affiliations:** 1State Key Laboratory of Freshwater Ecology and Biotechnology, Institute of Hydrobiology, Chinese Academy of Sciences, Wuhan 430072, China; dongzhang0725@gmail.com (D.Z.); zouhong@ihb.ac.cn (H.Z.); wusgz@ihb.ac.cn (S.G.W.); liming82101920@aliyun.com (M.L.); gtwang@ihb.ac.cn (G.T.W.); 2Key Laboratory of Aquaculture Disease Control, Ministry of Agriculture, Institute of Hydrobiology, Chinese Academy of Sciences, Wuhan 430072, China; 3University of Chinese Academy of Sciences, Beijing 100000, China; 4Bio-Transduction Lab, Wuhan 430075, China; ivanjakovlic@yahoo.com (I.J.); zhangjin2001@163.com (J.Z.); chenrong2S@163.com (R.C.)

**Keywords:** *Ancyrocephalus mogurndae*, positive selection, Monogenea, Dactylogyridae, flatworms, replication initiation, mtDNA

## Abstract

Whereas a majority of monogenean flatworms are ectoparasitic, i.e., parasitize on external surfaces (mainly gills) of their fish hosts, *Enterogyrus* species (subfamily Ancyrocephalinae) are mesoparasitic, i.e., parasitize in the stomach of the host. As there are numerous drastic differences between these two environments (including lower oxygen availability), we hypothesized that this life-history innovation might have produced adaptive pressures on the energy metabolism, which is partially encoded by the mitochondrial genome (OXPHOS). To test this hypothesis, we sequenced mitochondrial genomes of two Ancyrocephalinae species: mesoparasitic *E. malmbergi* and ectoparasitic *Ancyrocephalus mogurndae*. The mitogenomic architecture of *E. malmbergi* is mostly standard for monogeneans, but that of *A. mogurndae* exhibits some unique features: missing *trnL2* gene, very low AT content (60%), a non-canonical start codon of the *nad2* gene, and exceptionally long tandem-repeats in the non-coding region (253 bp). Phylogenetic analyses produced paraphyletic Ancyrocephalinae (with embedded Dactylogyrinae), but with low support values. Selective pressure (PAML and HYPHY) and protein structure analyses all found evidence for adaptive evolution in *cox2* and *cox3* genes of the mesoparasitic *E. malmbergi*. These findings tentatively support our hypothesis of adaptive evolution driven by life-history innovations in the mitogenome of this species. However, as only one stomach-inhabiting mesoparasitic monogenean was available for this analysis, our findings should be corroborated on a larger number of mesoparasitic monogeneans and by physiological studies.

## 1. Background

In contrast to a vast majority (more than 95% in estimation [1]) of other monogenean flatworms, which parasitize on external body surfaces, such as fins, skins, and gills of fish hosts, species belonging to the mesoparasitic genus *Enterogyrus* (Ancyrocephalinae, Dactylogyridae) parasitize in the digestive system (stomach) of their fish hosts [2,3]. This genus currently contains eight known species [4]. They insert the sharp curved terminal ends of their anchors into the epithelial tissue (stomach wall) [3,5]; thus, causing wounds and chronic morbidity to the host [5]. Their peculiar mesoparasitic lifestyle enables them to survive environmental changes associated with long inter-continental marine migrations (such as variations in salinity), which results in their vicariant global distribution, unlike that of ectoparasitic monogeneans [6,7]. However, it remains unclear which physiological and genetic adaptations facilitated this migration from gills and fins (plesiomorphic state for monogeneans [8]) to the drastically different, and arguably, inhospitable environment of the stomach, characterized by high acidity, absence of light, and lower oxygen availability, etc.

Mitochondrion is the energy factory in metazoans, by encoding the genes that make up four of the five subunits of the oxidative phosphorylation enzymatic complex (OXPHOS): Complex I (*nad1*–*6*), III (*cytb*), IV (*cox1–3*), and V (*atp6* and *atp8*) (Complex II is completely encoded by the nuclear genes) [9]. Due to the abundance of mitochondria in cells, (mostly) maternal inheritance, absence of introns, small genomic size (in metazoans), and an increasingly large set of available orthologous sequences, mitochondrial genomes (mitogenomes) have become a popular tool in population genetics [10], phylogenetics [11,12], and diagnostics [13] studies. An additional reason for their popularity in these types of studies is the fact that they were long considered to be an almost neutral marker, evolving under a strong purifying selection [14,15,16]. However, there is growing evidence for episodic positive selection in mitogenomes of some taxa, mostly identifiable as non-synonymous substitutions in second and first codon positions of OXPHOS genes [9,17,18,19]. In particular, radical changes in physicochemical properties of amino acids are usually seen as evidence for the signature of positive selection [17,19]. These amino acid (AA) substitutions can increase or decrease the aerobic capacity in response to the environmental adaption [18]. For example, Scott et al. [19] identified a mutation in the *cox3* gene of bar-headed goose that may have resulted in adaptations in mitochondrial enzyme kinetics and O_2_ transport capacity, and thereby enabled this species to fly at exceptionally high altitudes; Almeida et al. [9] found adaptive mitogenomic changes to the deep sea environment in cephalopods; Guo et al. [20] detected hypoxia adaptation in the mitogenomes of anchialine cave shrimps; and Yu et al. [21] found evidence of positive selection in the mitochondrial NADH dehydrogenase genes of Chinese snub-nosed monkeys that live in high-altitudes.

The energy metabolism, crucial for any living organism, is affected by the environmental conditions [22]. The examples above show that environmental changes (especially the concentration of oxygen) can increase the fitness of an organism carrying a beneficial mutation, which can be detected as signals of adaptive selection in mitochondrial OXPHOS genes. Adult helminths are generally considered to be facultative aerobes, and their oxygen consumption is in direct positive correlation with the availability of oxygen [23]. Hypoxic environment limits the efficiency of metabolic pathways that rely on acetyl-CoA as a substrate and decreases the energy output of mitochondria [24]. For example, some endoparasites of mammals adapt to the low oxygen availability in the digestive tract (which is about 1/4 of the air concentration) by switching to the anaerobic glycolytic phosphoenolpyruvate carboxykinase (PEPCK)-succinate pathway [25]. Although much less is known about metabolic adaptations of fish parasites, it is reasonable to assume that a switch from the ectoparasitic lifestyle (the gills environment being characterized by a relative abundance of oxygen and almost neutral PH) to the mesoparasitic lifestyle (the stomach environment being characterized by much lower availability of oxygen and highly acidic PH) must have been accompanied by comparable metabolic adaptations. Therefore, as the ancestor of the stomach-dwelling parasites from the genus *Enterogyrus* was almost certainly an ectoparasite (gills of fins) [8], this genus presents a good model to study signatures of selection in proteins encoded by the mitochondrial genome.

So far, such studies were hindered by the unavailability of mitogenomic data for the mesoparasitic monogeneans. Here, we sequenced and characterized two complete mitochondrial genomes belonging to two species from the subfamily Ancyrocephalinae with different life-histories, ectoparasitic (gills) *Ancyrocephalus mogurndae* (Yamaguti, 1940), and mesoparasitic (stomach) *E. malmbergi* Bilong Bilong, 1988, and searched for the evidence of adaptive selection signals in the latter mitogenome.

## 2. Materials and Methods

### 2.1. Specimen Collection and Identification

*A. mogurndae* was obtained from the gills of *Siniperca chuatsi* (Basilewsky, 1855) (Centrarchiformes: Sinipercidae) bought at a local market in the Wuhan city, Hubei Province on 17th of January 2017. *E. malmbergi* was obtained from the stomach of *Oreochromis niloticus* (Linnaeus, 1758) (Cichliformes: Cichlidae), caught in a freshwater pond in the Sun Yat-sen University (Guangzhou city, Guangdong province). Morphological identification of the latter species was conducted as described previously [3], so further details are not given here, whereas *A. mogurndae* was identified morphologically by the hard parts of the haptor and reproductive organs, as described in Wu et al. [26]. Additionally, to confirm the taxonomic identity of *A. mogurndae*, its *28S rRNA* gene was amplified using a primer pair designed before [27]: C1 (5′-ACCCGCTGAATTTAAGCAT-3′) and D2 (5′-TGGTCCGTGTTTCAAGAC-3′). It shares a very high identity of 99.74% with conspecific homologs available in GenBank: 779/781 identical bp with AY841871. All sampled and identified parasites were first washed in 0.6% saline and then stored in 100% ethanol at 4 °C.

### 2.2. DNA Extraction, Amplification, and Sequencing

Due to a small size of these parasites, to ensure a sufficient amount of DNA for amplification and sequencing, we used two kinds of genomic DNA: mixture DNA (20 parasite specimens) and individual DNA (a single parasite specimen). Both were extracted using TIANamp MicroDNA Kit (Tiangen Biotech, Beijing, China). The mixture DNA was first used to amplify the whole mitogenome. First, we selected 14 monogenean mitogenomes from the GenBank, aligned them using ClustalX [28], and designed degenerate primer pairs (Appendix A) matching the generally conserved regions of mitochondrial genes. On the basis of these obtained fragments, specific primers were designed using Primer Premier 5 [29], and the remaining mitogenome was amplified and sequenced in several PCR steps (Appendix A). PCR products were sequenced bi-directionally using both degenerate and specific primers mentioned above on an ABI 3730 automatic sequencer using the Sanger method. All obtained fragments were BLASTed [30] to confirm that the amplicon was the actual target sequence. To address the possibility of intraspecific sequence variation present in the mixture DNA, we inspected all chromatograms for double peaks or any other sign of the existence of two different sequences, and then we used individual DNA and long-range PCR to further verify the obtained sequences. If we found two different sequences, we used the DNA extracted from a single specimen to assemble the mitogenomic sequence, thereby ensuring that the entire sequence belongs to a single specimen.

### 2.3. Sequence Annotation and Analyses

Both mitogenomes were assembled and annotated following the procedure described before [31,32,33,34] using DNAstar v7.1 software [35], and MITOS [36] and ARWEN [37] web tools. The annotated genome was recorded in a Word document, and PhyloSuite software [38] was then used to parse and extract the annotation information, as well as create GenBank submission files and organization tables for mitogenomes. Data for the comparative genomics analyses were also extracted and processed using PhyloSuite. Tandem Repeats Finder [39] was employed to find tandem repeats in the long non-coding regions (LNCR), and their secondary structures were predicted by the Mfold software [40]. Genetic distances (identity) among mitogenomic sequences were calculated with the “DistanceCalculator” function in Biopython using the “identity” model.

### 2.4. Phylogenetic Analyses

Phylogenetic analyses were conducted using the two newly sequenced ancyrocephaline mitogenomes and 28 monogenean mitogenomes available in the GenBank (22/June/2019). Among these species, six polyopisthocotylean monogeneans were used as outgroups (Appendix A). Amino acid alignment of 12 protein-coding genes was used for the phylogenetic analyses. Best-fit homogeneous model (mtZOA+F+I+G4) was calculated with ModelFinder [41], integrated in PhyloSuite. On the basis of the selected model, phylogenetic analyses were performed using two different algorithms: maximum likelihood (ML) and Bayesian inference (BI). ML analysis was conducted using RAxML [42] using an ML+rapid bootstrap (BS) algorithm with 1000 replicates. Bayesian inference analysis with the empirical MTZOA model was conducted using PhyloBayes (PB) MPI 1.5a [43]. For each analysis, two MCMC chains were run after the removal of invariable sites from the alignment, and the analysis was stopped when the conditions considered to indicate a good run (PhyloBayes manual) were reached: maxdiff <0.1 and minimum effective size >300. Phylograms were visualized and annotated using iTOL [44], with the help of several dataset files generated by PhyloSuite.

### 2.5. Selection Analyses

Based on the tree reconstructed above, mutation pressures acting on the 12 mt OXPHOS genes were tested by comparing the nonsynonymous/synonymous substitution ratios (ω = dN/dS), through which positive (ω > 1), neutral (ω = 1), and purifying (ω < 1) selection can be distinguished. To avoid mutational saturation, which can blur the evolutionary signal at higher taxonomic levels, only monopisthocotyleans monogeneans, excluding Gyrodactylidae, were included in the selection analysis (15 species in total). First, we invoked the CODEML package in PAML 4.7 [45] to conduct a branch-site evolutionary analysis of the 12 PCGs. The modified branch-site model A allowed ω to vary, both among sites in the protein and across branches on the tree. After specifying foreground-lineages (*E. malmbergi*), the null model fixes ω to 1, whereas the alternative model assumes that sequences may have experienced positive selection (ω > 1). The significance of differences between the two models was assessed using likelihood ratio tests (LRTs). Subsequently, positively selected sites of genes with *p* < 0.05 in LRTs were further evaluated using the Bayes Empirical Bayes (BEB) [45] analysis with posterior probabilities ≥ 0.95. A mixed-effects model of evolution (MEME) and fixed-effect likelihood (FEL) methods implemented in HYPHY [46] were also used to infer the positively selected sites on the same foreground-lineage. Sites were considered as candidates under positive selection when they met the following conditions: β^+^ > α, significant likelihood ratio test (*p* < 0.05) in MEME, and *p* < 0.05 in the likelihood ratio test of FEL.

The optimal 3D structures of proteins were predicted using I-TASSER [47] on the basis of the confidence score (C-score) value. Superimposition, visualization, and manipulation of the 3D structures were conducted using PYMOL [48], and the prediction of protein structure and function using the PredictProtein server [49].

## 3. Results

### 3.1. Genome Organization and Base Composition

The complete mitochondrial genomes of *E. malmbergi* (MN095193) and *A. mogurndae* (MN095192) are circular molecules of 14,760 bp and 14,107 bp, respectively (Figure 1). The *E. malmbergi* mitogenome contains the standard [50] 36 flatworm mitochondrial genes (12 protein-encoding genes (PCGs; *atp8* is absent), 22 tRNA genes, and 2 rRNA genes), whereas we managed to identify only 35 genes in the mitogenome of *A. mogurndae*, from which the *trnL2* gene appears to be missing (Table 1 and Figure 1). The PCGs encoded by both mitogenomes mostly used canonical start and stop codons (start: ATG and GTG; stop: TAA and TAG), but the *nad2* gene of *A. mogurndae* uses non-canonical codons: the abbreviated version of TAA (T--) as the stop codon (otherwise relatively common, but unique in our dataset), and it proved difficult to determine the initial codon (Table 1 and Appendix A). The architecture and similarity of orthologous sequences of the two newly sequenced mitogenomes are summarized in Table 1. The average sequence similarity of PCGs between the two studied mitogenomes ranged from 43.14% (*nad4L*) to 66.57% (*cytb*) (Table 1). The gene order of *E. malmbergi* (Figure 1) was identical to that of *E. johnii*, *Tetrancistrum nebulosi*, *Cichlidogyrus sclerosus* and *Paragyrodactylus variegatus*, whereas the gene order of *A. mogurndae* was almost identical, with the exception of a transposition of *trnK* and the missing *trnL2*.

### 3.2. Non-Coding Regions

The long non-coding region (LNCR) was located between *nad5* and *trnG* genes in both mitogenomes (Figure 2). The two segments were 983 bp and 1430 bp in size in *E. malmbergi* and *A. mogurndae,* respectively. The LNCR of *E. malmbergi* contained two highly repetitive regions (HRR): HRR1 was composed of four successive tandem repeats (TRs), where repeat units 1–3 were identical (122 bp), whereas unit 4 lost one nucleotide at the end; HRR2 was comprised of 23 successively short TRs, in which units 1–22 were identical (16 bp), whereas the unit 23 was 3 bp shorter at the 3′ end (Figure 2). Similarly, we found one HRR in the LNCR of *A. mogurndae*; it contained five successive long TRs, where repeat units 1–3 were identical (253 bp), whereas unit 4 was longer, with one nucleotide insertion at the 102nd position, and unit 5 exhibited one nucleotide deletion at the 253rd position (Figure 2). The consensus repeat patterns of the HRR1 in *E. malmbergi* and the HRR in *A. mogurndae* are capable of forming stem-loop structures (Figure 2).

### 3.3. Phylogeny and Selection Analyses

BI and ML produced an identical topology (Figure 3), wherein *A. mogurndae* was resolved as the basal species within the Dactylogyridae, followed by *E. malmbergi*. The remaining dactylogyrids were divided into two sister-clades: (*E. johni*, *Dactylogyrus lamellatus*) and (*T. nebulosi*, (*C. mbirizei*, (*C. sclerosus*, *C. halli*))). However, the support values within the Dactylogyridae clade in the ML tree were very low (Figure 3). The subfamily Ancyrocephalinae was rendered paraphyletic by the embedded Dactylogyrinae species, *D. lamellatus*.

With the mesoparasitic *E. malmbergi* selected as the test branch within the phylogenetic tree described above (only the topology of monopisthocotylean monogeneans, excluding gyrodactylids, was used), branch-site specific analyses in PAML, and MEME and FEL analyses in HYPHY, all detected evidence for positive selection in this branch (Figure 3, Table 2, and Appendix A). Under the branch-site model (BSM) test, *cox1*, *cox2*, *cox3*, *nad1,* and *nad5* were identified as genes under positive selection (*p* < 0.05; Appendix A). However, BEB analyses only identified several codons in *cox2* and *cox3* genes as undergoing positive selection (*p*-value < 0.05). In comparison, the MEME method identified episodic diversifying positive selection (*p*-value < 0.05) in eleven PCGs (all PCGs except *nad1*) (Appendix A). The FEL method found evidence of pervasive positive diversifying selection in all 12 PCGs except for *nad3*, which only had 30 sites under negative selection (at *p* ≤ 0.05; Appendix A). Intriguingly, *nad1* was special again, with only one site under pervasive positive diversifying selection and 137 sites under negative selection (Appendix A).

As positively selected sites are more reliable when they are supported by two or more methods [51,52], only *cox2* and *cox3* genes were used in subsequent analyses of positive selection on proteins. Positively selected sites that were consistently identified by three methods (BSM, MEME, and FEL) were mapped onto the predicted 3D structures of Cox2 and Cox3 proteins of *E. malmbergi* (Figure 4). PredictProtein analysis showed that these positively selected sites were located in, or close to, the functional regions: 13 positively selected sites were located within a helical transmembrane region, and one positively selected site (codon 171 in *cox2*) was located within a protein-binding region (Figure 4, Table 2).

## 4. Discussion

Mitogenomic architecture of *E. malmbergi* is mostly standard for monogeneans, but the mitogenome of *A. mogurndae* possesses some intriguing features: the absence of *trnL2* gene, low AT content (the lowest among the Monogenea; Appendix A, Appendix A), exceptionally long tandem-repeats within the NCR, and non-canonical start/stop codons of the *nad2* gene. On the basis of results reported in other related species [12,53], as a working hypothesis, we propose ATA as the initial codon of *nad2* in *A. mogurndae*. The gene order shared between *E. malmbergi* and other four monogeneans were previously identified as the plesiomorphic gene order for the subphylum Neodermata [54]. This finding further confirms the hypothesis that identical gene orders found in phylogenetically distant lineages are an indicator of shared ancestry [55]. TRs with high repeat numbers and large size have been reported before in the subclass Monopisthocotylea [32,56], but with 253 bp, the TR in *A. mogurndae* is the longest reported so far. Along with a few previous studies [32,56], these findings consistently reject the hypothesis that monopisthocotylean monogeneans possess fewer and smaller (in size) TRs in the LNCR than polyopisthocotylean monogeneans [57]. Since the presence of tandem repeats forming a stable secondary structure is often associated with replication origin in mitochondria [12,58,59], it appears likely that the repeat regions found in the LNCR are embedded within the control region.

Results of our phylogenetic analyses are partially consistent with the relationships obtained using *18S rDNA* and *28S rDNA*, in which subfamilies Dactylogyrinae, Ancylodiscoidinae, and Pseudodactylogyrinae were embedded within the subfamily Ancyrocephalinae [60,61,62,63], thus rendering it paraphyletic. However, this result supports the argument that, regardless of the exact status and interrelationships of these subfamilies, all of them originated within the family Dactylogyridae [64]. As nuclear and mitogenomic data reject the monophyly of Ancyrocephalinae, and even the morphological data do not provide full support for it [64], we can conclude with some confidence that a taxonomic revision shall be needed with regard to this taxon. However, the resolution of our approach is too limited (small number of samples) to officially propose it here, so future studies should address this issue using a larger number of samples and combined datasets (ideally all three data types: nuclear, mitochondrial, and morphological).

Although only a small proportion of monogeneans are mesoparasitic (less than 5% in estimation), they have a wide host range (fishes, mammals, and amphibians), exhibit a wide range of parasitic lifestyles with respect to the exact location in the body where they parasitize (stomach, heart, esophagus, cloaca, urinary bladder, and inside the eyelid), and they form phylogenetically distant clades (different subclasses and families) [1,6]. This indicates multiple independent origins of the mesoparasitic lifestyle within the Monogenea. Due to limited availability of data, here, we only compared the selection pressures between ectoparasitic and stomach-inhabiting mesoparasitic monogeneans. Among the 15 species that we included in the selection analyses, 14 are ectoparasitic monogeneans found on the gills, skin, and fins of fish hosts, whereas *E. malmbergi* is a mesoparasitic monogenean found in the stomach of fish hosts. We hypothesized that starkly different environments of ectoparasitic and mesoparasitic monogeneans (especially the supposed reduced availability of oxygen in the stomach in comparison to the gills) would require metabolic adaptations, which might be reflected on the mitochondrial genome. Our analyses do show that dozens of codons of mitochondrial OXPHOS genes were subjected to strong positive selection pressures (Table 2 and Appendix A) in *E. malmbergi*. Particularly, several positive selection sites in *cox2* and *cox3* were consistently validated by three different methods (Table 2), which provides strong evidence that these genes may have sustained adaptive evolution in the mesoparasitic *E. malmbergi*. This was further supported by the evidence that many positively-selected codons were posited in, or close to, the predicted functional regions (helical transmembrane region and protein binding region) in the structure of mitochondrial OXPHOS proteins (Figure 4 and Table 2). *cox2* and *cox3*, both of which were confirmed to have undergone positive selection in *E. malmbergi* by three methods, belong to the multi-subunit OXPHOS enzyme complex IV [9]. As this complex is believed to consume a majority of the O_2_ [65], this finding is indicative of metabolic adaptations to a comparatively (in relation to gills) hypoxic environment in the stomach in *E. malmbergi*. However, it remains unclear whether this adaptation is a reflection of an enhanced capacity for oxygen usage or a switch to a predominantly anaerobic metabolism. Some of the genes in other enzymatic complexes were also predicted to exhibit positive selection by one or two methods: Complex I, which is the largest OXPHOS enzymatic complex, complex III, which transfers electrons, and complex V, which mainly synthesizes the ATP [66]. Therefore, there is some evidence for positive selection in all four mitochondrial OXPHOS complexes, which suggests life-history innovation-driven adaptive evolution in the mitogenome of *E. malmbergi*.

The interpretation of these findings is hampered by the scarcity of physiological studies on these parasitic species. Notably, the exact concentration of oxygen in the stomach of their fish hosts remains unknown, it is unclear whether mesoparasitic species employ aerobic or anaerobic metabolic pathways, and we do not know whether these parasites may be able to gain oxygen from the host’s tissues and body fluids. However, as the majority of parasites do not use the oxygen available within the host [25], we can tentatively reject the last hypothesis. Furthermore, in mammals, the oxygen content in the small intestine is about 25% of that in the environment (air) [25], and in humans, the gas in the stomach contains approximately 15% oxygen (as opposed to 21% oxygen in the air) [67]. Therefore we can also safely assume that oxygen availability is significantly different between gills (which are constantly flushed by water) and the stomach environment, which is isolated from the surrounding water in freshwater fish most of the time (apart from when they are ingesting food) [68,69]. It should be noted that *O. niloticus* is a euryhaline fish that ingests water in high salinity environments, which may increase the availability of oxygen in the stomach. However, studies in a closely related species *Tilapia mossambica*, which exhibits much higher salinity tolerance than *O. niloticus* [70], have shown that its ingestion of water is fairly low in a freshwater environment [71]. This implies that any stomach-dwelling parasite of freshwater populations would face limited accessibility of oxygen. Finally, as only one stomach-inhabiting mesoparasitic species was sequenced, we could not establish whether the amino-acid replacements observed in *E. malmbergi* are unique to this species or fixed in all species within this genus (all are mesoparasitic species).

## 5. Conclusions

The existence of several unique architectural characteristics of the mitogenome of *A. mogurndae* is intriguing and indicates that the sequencing of further mitogenomes belonging to this and closely related genera may be interesting from the aspect of the evolution of mitogenomic architecture in the Monogenea. With regards to the status of the subfamily Ancyrocephalinae, the addition of two new mitogenomes (now seven available in total) did not manage to stabilize the topology, and the subfamily was paraphyletic. Although this is partially in agreement with previous molecular studies, support values for the topology were low, so we cannot draw any conclusions with confidence. More data is needed to resolve the status of this subfamily. Finally, in agreement with our hypothesis, we found evidence for adaptive evolution in the mitogenome of *E. malmbergi*. Although we tentatively attributed it to the adaptation to the evolution from the ectoparasitic to stomach-inhabiting mesoparasitic lifestyle, where the hypoxic environment of the stomach required adaptive changes in some mitochondrial OXPHOS genes, it remains unclear whether this adaptation is a reflection of an enhanced capacity for oxygen usage or a switch to a predominantly anaerobic metabolism. Also, due to a lack of physiological studies on these species, we cannot exclude other, less obvious explanations for these adaptations. In order to investigate the adaptive evolution of stomach-inhabiting mesoparasitic monogeneans comprehensively, we will have to sequence more data from other mesoparasitic species and simultaneously study their physiological and genetic adaptations to the mesoparasitic lifestyle.

## Figures and Tables

**Figure 1 genes-10-00863-f001:**
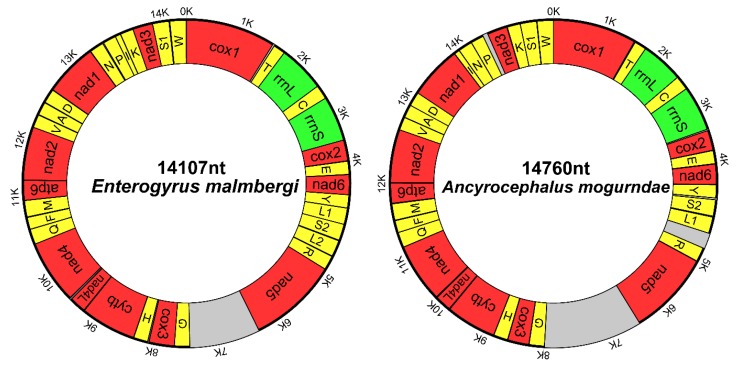
Visual representation of the circular mitochondrial genomes of *A. mogurndae* and *E. malmbergi*. Protein-coding genes are red, tRNAs are yellow, rRNAs are green, and non-coding regions are grey.

**Figure 2 genes-10-00863-f002:**
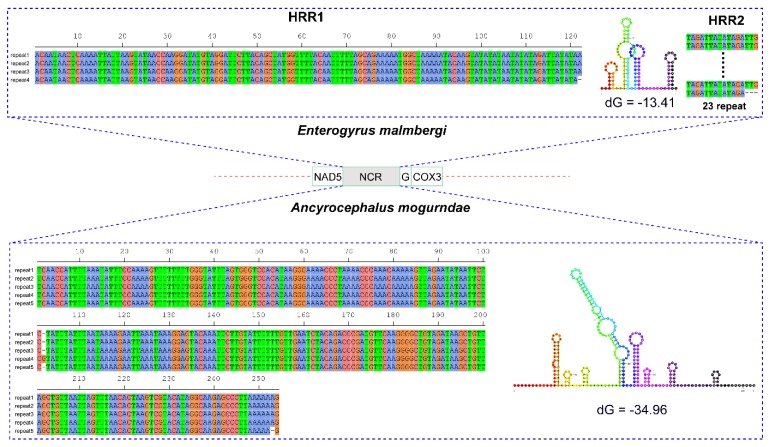
Stem-loop structures of the consensus repeat pattern in repetitive regions (RRs) of the non-coding region (NCR) of *A. mogurndae* and *E. malmbergi*. Thermodynamic energy values (dG) are shown next to the secondary structures. Only the NCR containing RRs was shown.

**Figure 3 genes-10-00863-f003:**
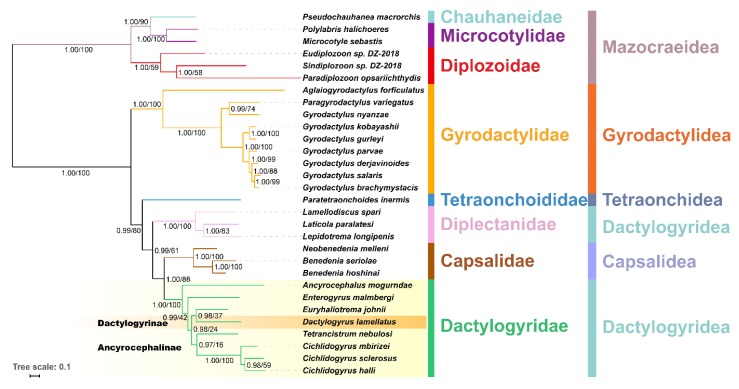
A phylogram reconstructed using mitogenomes of 30 monogeneans (amino acid sequences of all PCGs) and the mtZOA model. Scale bar corresponds to the estimated number of substitutions per site. Statistical support values of Bayesian analyses and maximum likelihood are shown on the nodes (left/right, respectively). Taxonomic families and orders are shown in different colors. Dactylogyrinae and Ancyrocephalinae subfamilies are also displayed.

**Figure 4 genes-10-00863-f004:**
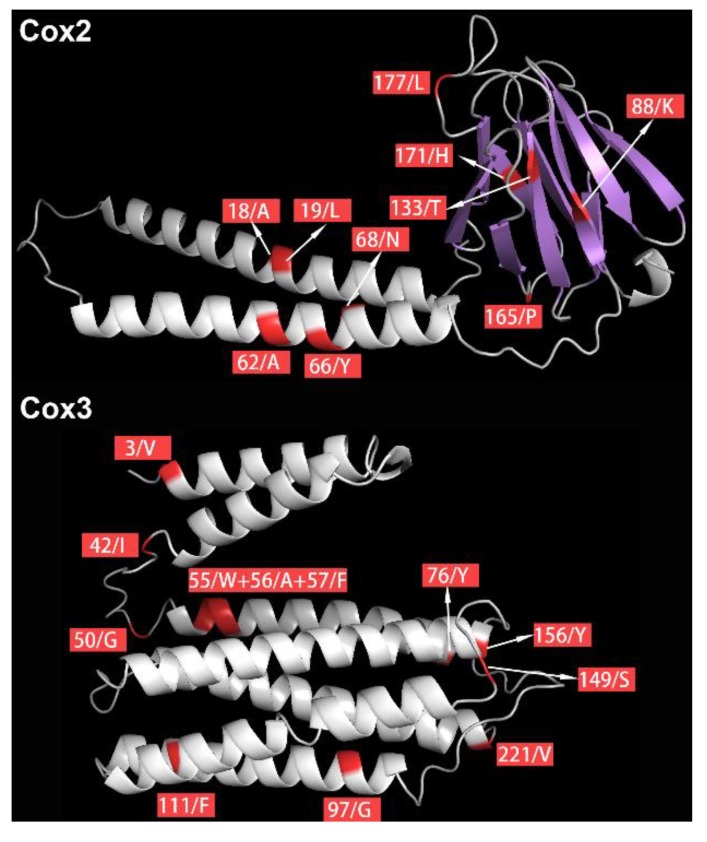
Distribution of positively selected sites in the three-dimensional (3D) structures of Cox2 and Cox3 proteins. The positively selected sites are shown in red.

**Table 1 genes-10-00863-t001:** Comparison of the annotated mitochondrial genomes of *Ancyrocephalus mogurndae* and *Enterogyrus malmbergi*.

Gene	Position		Size	Intergenic Nucleotides	Codon		Anti-codon	Identity
	From	To			Start	Stop		
*E. malmbergi*/*A. mogurndae*								
*cox1*	1/1	1551/1572	1551/1572		ATG/GTG	TAG/TAG		66.54
*trnT*	1579/1558	1646/1627	68/70	27/−15			TGT/TGT	57.14
*rrnL*	1647/1628	2599/2571	953/944					66.07
*trnC*	2600/2572	2665/2638	66/67				GCA/GCA	50
*rrnS*	2666/2639	3408/3357	743/719					60.64
*cox2*	3409/3379	4005/3951	597/573	0/21	ATG/ATG	TAA/TAA		55.02
*trnE*	4001/3947	4067/4014	67/68	−5/−5			TTC/TTC	47.89
*nad6*	4071/4022	4517/4468	447/447	3/7	ATG/GTG	TAA/TAG		50.56
*trnY*	4519/4476	4586/4542	68/67	1/7			GTA/GTA	47.83
*trnL1*	4585/4649	4650/4709	66/61	−2/14			TAG/TAG	44.93
*trnS2*	4651/4570	4716/4634	66/65	0/27			TGA/TGA	67.16
*trnL2*	4727/-	4794/-	68/-	10/0			TAA/-	
*trnR*	4802/4939	4865/5003	64/65	7/229			TCG/TCG	64.62
*nad5*	4867/5009	6420/6568	1554/1560	1/5	ATG/ATG	TAG/TAG		50.35
*trnG*	7404/7999	7469/8065	66/67	983/1430			TCC/TCC	56.72
*cox3*	7470/8070	8144/8723	675/654	0/4	ATG/ATG	TAA/TAG		50.81
*trnH*	8125/8719	8189/8782	65/64	−20/−5			GTG/GTG	69.23
*cytb*	8193/8786	9269/9871	1077/1086	3/3	ATG/ATG	TAA/TAG		66.57
*nad4L*	9278/9862	9529/10107	252/246	8/−10	ATG/ATG	TAG/TAG		43.14
*nad4*	9502/10095	10716/11309	1215/1215	−28/−13	ATG/ATG	TAG/TAG		51.18
*trnQ*	10717/11311	10783/11372	67/62	0/1			TTG/TTG	64.71
*trnF*	10782/11372	10846/11437	65/66	−2/−1			GAA/GAA	68.66
*trnM*	10839/11430	10904/11495	66/66	−8/−8			CAT/CAT	62.69
*atp6*	10908/11505	11417/12017	510/513	3/9	ATG/ATG	TAA/TAG		56.48
*nad2*	11422/12025	12243/12850	822/826	4/7	ATG/ATA	TAG/T		47.32
*trnV*	12244/12851	12308/12916	65/66				TAC/TAC	65.15
*trnA*	12316/12921	12377/12982	62/62	7/4			TGC/TGC	74.19
*trnD*	12385/12983	12447/13048	63/66	7/0			GTC/GTC	43.94
*nad1*	12453/13049	13340/13933	888/885	5/0	ATG/ATG	TAG/TAG		63.18
*trnN*	13346/14006	13405/14073	60/68	5/2			GTT/GTT	60
*trnP*	13421/14076	13487/14141	67/66	15/2			TGG/TGG	62.69
*trnI*	13487/13937	13554/14003	68/67	−1/3			GAT/GAT	82.35
*trnK*	13558/14568	13622/14634	65/67	3/7			CTT/CTT	54.41
*nad3*	13624/14216	13971/14560	348/345	1/74	ATG/ATG	TAG/TAG		51.13
*trnS1*	13971/14634	14029/14691	59/58	−1/−1			GCT/TCT	67.8
*trnW*	14042/14699	14104/14760	63/62	12/7			TCA/TCA	61.9

**Table 2 genes-10-00863-t002:** Positively selected sites of *cox2* and *cox3* genes identified by PAML, and HYPHY.

GENES	AA Positions	PAML Branch-Site Model (*p*-Value < 0.05, Posterior Probabilities ≥ 95%)	HYPHY		PredictProtein (https://www.predictprotein.org//)
			FEL *p* < 0.05	MEME *p* < 0.05 & β^+^ > α	
*cox2*	2		✓	✓	
	5		✓	✓	Protein binding region
	6		✓	✓	Protein binding region
	7		✓		
	8		✓		Protein binding region
	9		✓	✓	Protein binding region
	11		✓		Helical transmembrane region
	15		✓	✓	Helical transmembrane region
	18	✓	✓	✓	Helical transmembrane region
	19	✓	✓	✓	Helical transmembrane region
	20		✓	✓	Helical transmembrane region
	28		✓		Helical transmembrane region
	31		✓	✓	Helical transmembrane region
	32		✓		Helical transmembrane region
	39		✓	✓	
	44		✓	✓	
	45	✓	✓		Protein binding region
	46		✓		
	47		✓	✓	
	49		✓		Helical transmembrane region
	51		✓		Helical transmembrane region
	53		✓	✓	Helical transmembrane region
	54		✓		Helical transmembrane region
	57		✓		Helical transmembrane region
	60		✓	✓	Helical transmembrane region
	61		✓		Helical transmembrane region
	62	✓	✓	✓	Helical transmembrane region
	64		✓	✓	Helical transmembrane region
	66	✓	✓	✓	Helical transmembrane region
	68	✓	✓	✓	Helical transmembrane region
	69		✓		Helical transmembrane region
	71		✓		
	72		✓	✓	
	73		✓	✓	
	74		✓	✓	
	79	✓	✓		Protein binding region
	80	✓	✓		Protein binding region
	81		✓		Protein binding region
	82		✓	✓	
	88	✓	✓	✓	
	91		✓		
	99		✓		
	117		✓		
	118		✓		
	119		✓		
	126		✓	✓	
	128		✓		
	129		✓		
	133	✓	✓	✓	
	136		✓		
	148		✓	✓	
	152		✓		
	154		✓		
	165	✓	✓	✓	
	170		✓		
	171	✓	✓	✓	Protein binding region
	177	✓	✓	✓	
*cox3*	2		✓	✓	Helical transmembrane region
	3	✓	✓	✓	Helical transmembrane region
	8		✓	✓	Helical transmembrane region
	11		✓		Helical transmembrane region
	15		✓	✓	Helical transmembrane region
	21		✓		
	25		✓		Helical transmembrane region
	40		✓	✓	Helical transmembrane region
	42	✓	✓	✓	
	44		✓	✓	
	47		✓	✓	
	49		✓		
	50	✓	✓	✓	
	52		✓		Protein binding region; Polynucelotide-binding region
	53		✓	✓	
	55	✓	✓	✓	Helical transmembrane region
	56	✓	✓	✓	Helical transmembrane region
	57	✓	✓	✓	Helical transmembrane region
	60		✓		Helical transmembrane region
	66		✓	✓	Helical transmembrane region
	71		✓		Helical transmembrane region
	73		✓	✓	Helical transmembrane region
	74		✓		Helical transmembrane region
	76	✓	✓	✓	Helical transmembrane region
	77		✓	✓	Protein binding region
	82		✓	✓	
	83		✓		
	85		✓		
	90		✓		
	91		✓		
	94		✓	✓	Helical transmembrane region
	96		✓		Helical transmembrane region
	97	✓	✓	✓	Helical transmembrane region
	101		✓	✓	Helical transmembrane region
	104		✓	✓	Helical transmembrane region
	111	✓	✓	✓	Helical transmembrane region
	112		✓	✓	Helical transmembrane region
	115		✓	✓	
	130		✓	✓	Helical transmembrane region
	131		✓		Helical transmembrane region
	132		✓	✓	Helical transmembrane region
	139		✓	✓	Helical transmembrane region
	147		✓		
	149	✓	✓	✓	
	150		✓		
	154		✓		
	155		✓		
	156	✓	✓	✓	
	157		✓	✓	
	160		✓	✓	Helical transmembrane region
	163		✓	✓	Helical transmembrane region
	164		✓		Helical transmembrane region
	165		✓	✓	Helical transmembrane region
	171		✓	✓	Helical transmembrane region
	175		✓	✓	Helical transmembrane region
	179		✓	✓	Helical transmembrane region
	180		✓	✓	Helical transmembrane region
	184		✓	✓	
	185		✓	✓	
	187		✓	✓	Protein binding region
	189		✓		Protein binding region
	191		✓		
	195		✓	✓	
	196		✓	✓	
	197		✓	✓	
	199		✓	✓	Helical transmembrane region
	216		✓		Helical transmembrane region
	217		✓	✓	Helical transmembrane region
	218		✓	✓	Helical transmembrane region
	220		✓	✓	Helical transmembrane region; Protein binding region
	221	✓	✓	✓	Helical transmembrane region

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
