# Peer review of "Evidence for Adaptive Selection in the Mitogenome of a Mesoparasitic Monogenean Flatworm Enterogyrus malmbergi"

_genes, 2019, doi:10.3390/genes10110863_

Round 1

Reviewer 1 Report

The aim of this research is to compare the mitogenome in two related species (i.e. belonging to the same family) which have very different ecologies, one being an ectoparasite on the gills, one being a mesoparasite in the stomach. Both species thus encounter different ecological conditions, including drastic differences in oxygen level. This is an excellent idea, and this makes this paper more interesting that many mitogenome descriptions.

I found the paper relatively well written, with sometimes sentences which need to be edited by a native English speaker – this probably concerns the title itself.

My corrections and suggestions concern mainly the part about systematics and phylogeny.

Name of various species.

Line 80 and possibly others. Enterogyrus malmbergi was described under this genus name, so the complete name with taxon author is: Enterogyrus malmbergi Bilong Bilong, 1988, NOT Enterogyrus malmbergi (Bilong Bilong, 1988) – no parentheses here. Correct this to respect the International Code of Zoological Nomenclature.

Line 79: Ancyrocephalus mogurndae (Yamaguti, 1940) (with parentheses as such) is correct.

Figures, etc. Pseudorhabdosynochus yangjiangensis is a junior synonym of Laticola paralatesi (Nagibina, 1976) Yang et al., 2006. The authors can consult Prof. Tingbao Yang about this, since he provided the samples (line 326). It is possible that the sequence of this species was described under an incorrect taxonomical name, but it is the duty of the authors to use the correct name: Laticola paralatesi. References:

Yang, T. B., Kritsky, D. C., Sun, Y., Zhang, J. Y., Shi, S. H., & Agrawal, N. (2006). Diplectanids infesting the gills of the barramundi Lates calcarifer (Bloch) (Perciformes: Centropomidae), with the proposal of Laticola n. g. (Monogenoidea: Diplectanidae). Systematic Parasitology, 63(2), 127-141.

Sigura, A., & Justine, J.-L. (2008). Monogeneans of the speckled blue grouper, Epinephelus cyanopodus (Perciformes, Serranidae), from off New Caledonia, with a description of four new species of Pseudorhabdosynochus and one new species of Laticola (Monogenea: Diplectanidae), and evidence of monogenean faunal changes according to the size of fish. Zootaxa, 1695, 1-44.

Figures, etc. Microcotyle sebastis. The mitogenome of this species was described under this name and is registered as such in GenBank, but there are strong doubts about the real identity of the specimens used in this study. See:

Ayadi, Z. E. M., Gey, D., Justine, J.-L., & Tazerouti, F. (2017). A new species of Microcotyle (Monogenea: Microcotylidae) from Scorpaena notata (Teleostei: Scorpaenidae) in the Mediterranean Sea. Parasitology International, 66(2), 37-42. http://doi.org/10.1016/j.parint.2016.11.004

Name of higher taxa.

The authors use several times the word “polyopisthocotylid” and “monopisthocotylid”. While it is obvious by the text that they refer, respectively, to the Polyopisthocotylea and the Monopisthocotylea, the words they use are incorrect and are never used in good papers about monogeneans. The word  “polyopisthocotylid” looks like the anglicized version of a family Polyopisthocotylidae (Latin) which does not exist (same for “monopisthocotylids”). The authors should use “Polyopisthocotylea” and “Monopisthocotylea”, or “polyopisthocotylean monogeneans” and “monopisthocotylean monogeneans”. Line 246 is correct, but “subclass” might be deleted.

Note that “gyrodactylids”, used line 145, is correct – there is a family Gyrodactylidae.

Phylogenetic analysis

It seems that the authors do not have a very clear idea of the hierarchy of families and subfamilies. Subfamilies (ending in –inae) are PART of families (ending in –idae). Therefore, lines 256, it is not possible to have a subfamily “originated within the family”. The whole paragraph lines 252-262 needs rewriting; what is “this taxon”?

Apart from this minor problem, I should say that the status of the Ancyrocephalinae and Dactylogyridae has been disputed in many morphological and molecular papers over the years – the paper adds some information but it is clear that these taxa need revision.

Details

The sentence lines 64-68 has several parts, the three firsts begin by the name of the authors and the last one begin by the name of the species – please make parallel sentences.

Author Response

I found the paper relatively well written, with sometimes sentences which need to be edited by a native English speaker – this probably concerns the title itself.

RESPONSE: We proofread the entire paper again and corrected several grammatical errors. We also changed the title.

Line 80 and possibly others. Enterogyrus malmbergi was described under this genus name, so the complete name with taxon author is: Enterogyrus malmbergi Bilong Bilong, 1988, NOT Enterogyrus malmbergi (Bilong Bilong, 1988) – no parentheses here. Correct this to respect the International Code of Zoological Nomenclature.

RESPONSE: Thanks, done.

Figures, etc. Pseudorhabdosynochus yangjiangensis is a junior synonym of Laticola paralatesi (Nagibina, 1976) Yang et al., 2006. The authors can consult Prof. Tingbao Yang about this, since he provided the samples (line 326). It is possible that the sequence of this species was described under an incorrect taxonomical name, but it is the duty of the authors to use the correct name: Laticola paralatesi. References:

Yang, T. B., Kritsky, D. C., Sun, Y., Zhang, J. Y., Shi, S. H., & Agrawal, N. (2006). Diplectanids infesting the gills of the barramundi Lates calcarifer (Bloch) (Perciformes: Centropomidae), with the proposal of Laticola n. g. (Monogenoidea: Diplectanidae). Systematic Parasitology, 63(2), 127-141.

Sigura, A., & Justine, J.-L. (2008). Monogeneans of the speckled blue grouper, Epinephelus cyanopodus (Perciformes, Serranidae), from off New Caledonia, with a description of four new species of Pseudorhabdosynochus and one new species of Laticola (Monogenea: Diplectanidae), and evidence of monogenean faunal changes according to the size of fish. Zootaxa, 1695, 1-44.

RESPONSE: Thanks for spotting this mistake. We changed “Pseudorhabdosynochus yangjiangensis” to “Laticola paralatesi” in figures and supplementary files.

Figures, etc. Microcotyle sebastis. The mitogenome of this species was described under this name and is registered as such in GenBank, but there are strong doubts about the real identity of the specimens used in this study. See:

Ayadi, Z. E. M., Gey, D., Justine, J.-L., & Tazerouti, F. (2017). A new species of Microcotyle (Monogenea: Microcotylidae) from Scorpaena notata (Teleostei: Scorpaenidae) in the Mediterranean Sea. Parasitology International, 66(2), 37-42. http://doi.org/10.1016/j.parint.2016.11.004

RESPONSE: I have read this paper, and I agree about the uncertainty of the identity of the specimen. However, as the author doesn’t propose a clear solution, we decided to go with Microcotyle sebastis, and just add a note to the Supplementary Table S1, highlighting that the identity of this mitogenome is uncertain, and supported it with the above reference.

Name of higher taxa.

The authors use several times the word “polyopisthocotylid” and “monopisthocotylid”. While it is obvious by the text that they refer, respectively, to the Polyopisthocotylea and the Monopisthocotylea, the words they use are incorrect and are never used in good papers about monogeneans. The word  “polyopisthocotylid” looks like the anglicized version of a family Polyopisthocotylidae (Latin) which does not exist (same for “monopisthocotylids”). The authors should use “Polyopisthocotylea” and “Monopisthocotylea”, or “polyopisthocotylean monogeneans” and “monopisthocotylean monogeneans”. Line 246 is correct, but “subclass” might be deleted.

Note that “gyrodactylids”, used line 145, is correct – there is a family Gyrodactylidae.

RESPONSE: We changed the anglicized names throughout the text according to your objections.

Phylogenetic analysis

It seems that the authors do not have a very clear idea of the hierarchy of families and subfamilies. Subfamilies (ending in –inae) are PART of families (ending in –idae). Therefore, lines 256, it is not possible to have a subfamily “originated within the family”. The whole paragraph lines 252-262 needs rewriting; what is “this taxon”?

RESPONSE: We are not following your logic here. Subfamily is a subset of Family, and therefore, subfamilies can, and do, originate within families. Maybe you mixed up subfamilies and superfamilies, which indeed cannot originate within a family, but end with –idea? We left the text as it is, as we cannot spot any logical flaws there. We did slightly rewrite the paragraph, and provided additional clarifications where text was ambiguous (i.e. replaced ‘this taxon’ with Ancyrocephalinae).

The sentence lines 64-68 has several parts, the three firsts begin by the name of the authors and the last one begin by the name of the species – please make parallel sentences.

RESPONSE: Changed according to your objections: “…and Yu, et al. [21] found evidence of positive selection in the mitochondrial NADH dehydrogenase genes of Chinese snub-nosed monkeys that live in high-altitudes.”

Reviewer 2 Report

The authors have sequenced and analysed two mitochondrial genomes from two flatworms one with an ectoparasitic lifestyle Ancyrocephalus mogurndae and one mesoparasitic parasite Enteropyrus malmbergi. The authors hypothesise that the change from an ectoparasite on the gills to a stomach parasite would be accompanied by genetic mutations that should support a more anaerobic life style.

Overall the manuscript is well written however I have two general issues with this study.

It remains unclear to me what the oxygen levels in the stomach of the fish are? Are they significantly reduced?, Is the parasite maybe able to gain Oxygen from the host system? Are the positively selected amino acids in the subunits of COX2 and COX3 involved in the function of the two proteins. There is ample data on mutations of these two very conserved proteins. The authors should at least go into the literature and find examples that would support/disprove their idea that these mutations are involved in O2 consumption. Furthermore it is unclear to me what the authors mean by: “….E. malmbergi may have evolved an enhanced capacity for oxygen usage” Do they mean it is more efficiently using Oxygen? The authors should really read the related hypoxia literature and then come up with a plausible explanation

Author Response

The authors have sequenced and analysed two mitochondrial genomes from two flatworms one with an ectoparasitic lifestyle Ancyrocephalus mogurndae and one mesoparasitic parasite Enteropyrus malmbergi. The authors hypothesise that the change from an ectoparasite on the gills to a stomach parasite would be accompanied by genetic mutations that should support a more anaerobic life style.

Overall the manuscript is well written however I have two general issues with this study.

It remains unclear to me what the oxygen levels in the stomach of the fish are? Are they significantly reduced?, Is the parasite maybe able to gain Oxygen from the host system? Are the positively selected amino acids in the subunits of COX2 and COX3 involved in the function of the two proteins. There is ample data on mutations of these two very conserved proteins. The authors should at least go into the literature and find examples that would support/disprove their idea that these mutations are involved in O2 consumption. Furthermore it is unclear to me what the authors mean by: “….E. malmbergi may have evolved an enhanced capacity for oxygen usage” Do they mean it is more efficiently using Oxygen? The authors should really read the related hypoxia literature and then come up with a plausible explanation

RESPONSE: Regarding the first two questions, we failed to find related references. However, in comparison to the gills, which are constantly being flushed by water, the stomach of freshwater fish is isolated from the surrounding water most of the time (apart from when they are ingesting food) [1,2]. The situation may be a bit different in sea fish, which have to ingest water regularly [3,4], and some freshwater fish species that can ingest (gulp) air. Although O. niloticus is a euryhaline fish that ingests water in high salinity environments, which may increase the availability of oxygen in the stomach, our samples were collected from a freshwater pond. Furthermore, studies in a closely related species Tilapia mossambica, which exhibits much higher salinity tolerance than O. niloticus [5], have shown that its ingestion of water is fairly low in a freshwater environment [6]. This implies that any stomach-dwelling parasite of freshwater populations would face a limited accessibility of oxygen. Furthermore, we know that in mammals the oxygen content in small intestine is about 25% of that in the environment (air) [7], and that in humans the gas in the stomach contains approximately 15 % oxygen (as opposed to 21% oxygen in the air) [7]. Therefore, although we don’t know the exact oxygen concentration in the stomach, we cannot imagine any scenario that would invalidate our working hypothesis of the gill-to-stomach lifestyle switch requiring notable physiological adaptations. Regarding the question “Is the parasite maybe able to gain Oxygen from the host system?”, we found a reference that states “The majority of parasites do not use the oxygen available within the host” [7], but this still needs further validation. Regarding the remaining questions, functional studies on these proteins are mostly limited to model mammal species. Although we did conduct protein 3D modelling, due to absence of suitable (i.e. phylogenetically closely related) models in the PDB, these results lack precision necessary to discuss specific functional impacts on these proteins. As we have mentioned in the MS, cox2 and cox3 belong to multi-subunit OXPHOS enzyme complex IV, and this complex is believed to consume a majority of the O2. Additionally, several positive selection sites in cox2 and cox3 were consistently validated by three different methods, and they were posited in, or close to, the predicted functional regions (helical transmembrane region and protein binding region) in cox2 and cox3. Therefore, these mutations may change the function of the genes, probably involved in O2 consumption. Additionally, it is possible that these mutations may attribute to the adaptive evolution of the change of the habitat. Regarding the question of “….E. malmbergi may have evolved an enhanced capacity for oxygen usage”, we have changed the sentence to “However, it remains unclear whether this adaptation is a reflection of an enhanced capacity for oxygen usage or a switch to a predominantly anaerobic metabolism.”. We added an entire paragraph discussing the limitations of our study at the end of the Discussion section and modified the contents where needed (Abstract, Background and Conclusions).

REFERENCES

Smith, M.W. The in vitro absorption of water and solutes from the intestine of goldfish, Carassius auratus. J. Physiol. 1964, 175, 38-49. Ern, R.; Huong, D.T.T.; Cong, N.V.; Bayley, M.; Wang, T. Effect of salinity on oxygen consumption in fishes: a review. J. Fish Biol. 2014, 84, 1210-1220. Taylor, J.R.; Grosell, M. The intestinal response to feeding in seawater gulf toadfish, Opsanus beta, includes elevated base secretion and increased epithelial oxygen consumption. J. Exp. Biol. 2009, 212, 3873-3881. Skadhauge, E.; Lotan, R. Drinking rate and oxygen consumption in the euryhaline teleost Aphanius dispar in waters of high salinity. J. Exp. Biol. 1974, 60, 547-556. Yamaguchi, Y.; Breves, J.P.; Haws, M.C.; Lerner, D.T.; Grau, E.G.; Seale, A.P. Acute salinity tolerance and the control of two prolactins and their receptors in the Nile tilapia (Oreochromis niloticus) and Mozambique tilapia (O. mossambicus): A comparative study. Gen. Comp. Endocrinol. 2018, 257, 168-176. Potts, W.; Foster, M.; Rudy, P.; Howells, G.P. Sodium and water balance in the cichlid teleost, Tilapia mossambica. J. Exp. Biol. 1967, 47, 461-470. Kita, K.; Hirawake, H.; Miyadera, H.; Amino, H.; Takeo, S. Role of complex II in anaerobic respiration of the parasite mitochondria from Ascaris suum and Plasmodium falciparum. Biochim. Biophys. Acta Bioenerg. 2002, 1553, 123-139.

Round 2

Reviewer 2 Report

Adding the section in the discussion improves the manuscript significantly. This should also be reflected in the title....clearly the study does not reveal evidence for adaptation to hypoxia...Please adapt the title accordingly.

Author Response

Adding the section in the discussion improves the manuscript significantly. This should also be reflected in the title....clearly the study does not reveal evidence for adaptation to hypoxia...Please adapt the title accordingly.

RESPONSE: We find this question confusing. Especially the part “clearly the study does not reveal evidence for adaptation to hypoxia “. Please note that the old title referred to us finding evidence for hypoxia: "Mitochondrial genomes of ectoparasitic Ancyrocephalus mogurndae and mesoparasitic Enterogyrus malmbergi (Monogenea: Ancyrocephalinae) reveal the evidence for adaptation to hypoxia in the latter". However, in the first revision, we removed the word hypoxia from it, and thoroughly reorganised the title: "Evidence for adaptive selection in the mitogenome of a mesoparasitic monogenean flatworm Enterogyrus malmbergi". We are confident that this title correctly reflects our findings, as we did find strong evidence for adaptive selection. However, as we are not absolutely sure that it is hypoxia that drives the adaptation, we removed that term from the title during the first revision, and merely stated that the mitogenome exhibits signs of adaptive selection. Although we do mention that this parasite is mesoparasitic, and in the paper we hypothesise that adaptive changes are driven by this life history adaptation, we also cannot conclude this with confidence, as the adaptation may have hypothetically been driven by other factors, overseen by us. Therefore, we also avoided explicitly making this connection in the title, i.e., we made sure that we don't explicitly state that mitogenome exhibits adaptation to the mesoparasitic lifestyle. Therefore, we feel that we have very carefully avoided making any statements that are not fully supported by our data. As the reviewer states that the title should be changed because “…the study does not reveal evidence for adaptation to hypoxia”, our guess is that reviewer probably did not notice that we changed the title during the revision, and instead still referred to the old title.